# Polymeric Encapsulate of *Streptomyces* Mycelium Resistant to Dehydration with Air Flow at Room Temperature

**DOI:** 10.3390/polym15010207

**Published:** 2022-12-31

**Authors:** María Elena Mancera-López, Josefina Barrera-Cortés, Roberto Mendoza-Serna, Armando Ariza-Castolo, Rosa Santillan

**Affiliations:** 1Biotechnology and Bioengineering Department, Center for Research and Advanced Studies of the National Polytechnic Institute, Zacatenco Unit, Mexico City 07360, Mexico; 2Career of Chemical Engineering, Multidisciplinary Experimental Research Unit (UMIEZ), Faculty of Higher Studies Zaragoza, National Autonomous University of Mexico, Mexico City 09230, Mexico; 3Chemistry Department, Center for Research and Advanced Studies of the National Polytechnic Institute, Zacatenco Unit, Mexico City 07360, Mexico

**Keywords:** actinomycetes, alginate capsules, drying treatment, rotary drum, vegetative cells

## Abstract

Encapsulation is one of the technologies applied for the formulation of biological control agents. The function of the encapsulating matrix is to protect the biological material from environmental factors, while dehydration allows for its viability to be prolonged. An advantage of dehydrated encapsulation formulations is that they can be stored for long periods. However, vegetative cells require low-stress dehydration processes to prevent their loss of viability. Herein we describe the fabrication of a dehydrated encapsulate of the *Streptomyces* CDBB1232 mycelium using sodium alginate with a high concentration of mannuronic acid; sodium alginate was added with YGM medium for mycelium protection purposes. The encapsulation was carried out by extrusion, and its dehydration was carried out in a rotating drum fed with air at room temperature (2–10 L min^−1^). The drying of the capsules under air flows higher than 4 L min^−1^ led to viability loss of the mycelium. The viability loss can be decreased up to 13% by covering the alginate capsules with gum arabic. Compared to conventional dehydration processes, air moisture removal can be lengthy, but it is a low-cost method with the potential to be scaled.

## 1. Introduction

*Streptomyces* are Gram-positive filamentous bacteria that are commonly found in the rhizosphere of plants or in their roots. This bacteria genus has a high potential to produce secondary metabolites, which have been extensively researched in the medical field [1]. Approximately two-thirds of clinical antibiotics and other bioactive compounds are synthesized with bacteria of the *Streptomyces* genus. In the agricultural sector, the study of *Streptomyces* has focused on applications of biological control and stimulation of plant growth [2]. In the area of biological control, bacteria of the *Streptomyces* genus have shown properties antagonistic to fungi that colonize the root system of plants [3,4]. Biocontrol is carried out by secondary metabolites (e.g., polyoxin D, streptomycin, and kasugamycin) and extracellular enzymes that affect the growth of pathogens [1,2,3,4]. The plant growth is stimulated by the better solubilization of the nutrients contained in the soil and the availability of growth hormones [5,6,7].

Despite the importance of the *Streptomyces* bacteria in agricultural crops, only a few phytosanitary products containing this microorganism genus are available: *Streptomyces griseoviridis* K61 (Mycostop^®^, Finland), *Streptomyces lydicus* WYEC 108 (Actinovate^®^, Micro108^®^, Action Iron^®^, USA), and *Streptomyces saraceticus* KH400 (YAN TEN *Streptomyces saraceticus*, Taiwan), of which Actinovate^®^ and Mycostop^®^ are the best known and used for the treatment of agricultural soils [8,9]. The active agents in these products are mainly spores formulated as wettable powders or granules. Spores are the resistant phase of many microorganisms and have the property of resisting high shear stresses and extreme environmental conditions [10,11]. Vegetative cells are less resistant than spores; however, their incorporation in dehydrated solid formulations is of interest considering that not all microorganisms sporulate and they are an already viable phase with a high probability of propagation (Appendix A) [12,13,14,15,16].

Dehydration is the technology commonly applied for the conservation of microbial strains [14,17,18]. Dehydration (or anhydrobiosis) is a moisture removal process that aims to reduce or stop the vital functions of microorganisms until they reach a state of lethargy; the microorganisms reestablish their metabolism when rehydrated [19]. Additional advantages of dehydrated formulations (powdered or granulated) are the cost reduction of storage and refrigeration and their lower propensity for contamination [20]. The disadvantage is the viability loss of strains, whose magnitude depends on the strain characteristics (age of the microorganism, cell concentration, and strain tolerance to dehydration), as well as on the dehydration method applied (Appendix A) [14,21,22,23,24,25,26,27,28,29,30]. In the case of vegetative cells, lyophilization and cryopreservation are the technologies commonly applied, but mainly for the purpose of strain preservation [14,18]. When larger volumes of biomass must be dehydrated, heating or moisture removal with hot air are the preferred alternative technologies because of the low cost of the involved operations [31]. The inconvenience of moisture removal with heat is the viability loss of the microbial cells because of their high sensitivity to heat or shear stresses, which usually damage the cellular membrane. Thus, the encapsulation of the biological material prior to dehydration is the strategy of interest and under study, considering the great variety of solid supports available (inorganic, organic, and polymeric) and the possibility of modifying their physicochemical properties by integrating into them the appropriate additive or additives (Appendix A) [4,14,32,33,34,35,36,37,38].

Of the known polymeric supports, alginate is one of the most commonly used in the encapsulation of microbial strains [34,35,36,37]. Alginate is a biocompatible, biodegradable, and nontoxic support that can be applied to encapsulate hydrophilic active substances. Mechanical and chemical stability are the desired properties of an encapsulating support to allow preservation of the metabolic activity of the microbial agents [37]. In the case of alginates, these properties are determined by the mannuronic and guluronic acid concentrations that integrate it, as well as by the crosslinking agent used to form the gel [37]. A mechanically unstable alginate gel can be strengthened by incorporating on it selected additives or by covering the alginate capsules with alginate-compatible polymeric materials [4,33,38]. Calcium alginate is a highly hygroscopic support. However, its high porosity can favor the moisture removal. The high porosity of the alginate may favor the development of fractures during its drying with heat. However, as mentioned above, the fractures can be eliminated if additives are incorporated into the alginate matrix. In the drying of alginate films (gelatin-sodium alginate edible films), Al-Harrasi et al. (2022) [39] reported that humidity removal at room temperature allows for maintaining a uniform film surface without cracks, compared to those dried at 45 °C. However, the drawback of this type of drying is the time, which can be prolonged depending on the air humidity coming into the drier, as well as the resistance to mass transfer offered by the encapsulated biological material [31]. The aim of this work was to study the drying process of a *Streptomyces* culture encapsulated in a calcium alginate matrix. *Streptomyces* was propagated in liquid culture, so the encapsulated biological material comprised pellets of 2–5 mm in diameter. Fragmented mycelium formulations are more common; however, the pellet mycelium is assumed to be more resistant to shear and stress conditions. Air-drying of encapsulated strains of *Streptomyces* has generated losses of viability in a magnitude of 4–5 powers, even immobilizing the vegetative cells prior to encapsulation in different polymeric materials (Appendix A). In the present work, the encapsulation of the *Streptomyces* mycelium and the coating of the capsules with gum arabic prior to their dehydration in a rotary drum dryer allowed the dehydration of the biological material with low viability losses. To our knowledge, the use of such systems to dehydrate the mycelium has not been previously reported. In addition, given the characteristics of the designed drying system, it can be applied to the drying of microorganisms of different structures (spores, mycelium, endospores, etc.), and its operation is low-cost.

## 2. Materials and Methods

### 2.1. Microorganisms and Culture Conditions

*Streptomyces* (CDBB1232) was obtained from the National Collection of Microbial and Cell Cultures (CDBB, Mexico City, Mexico) of the Center for Research and Advanced Studies of the National Polytechnic Institute (CINVESTAV-IPN). Strain samples of 100 μL were seeded in triplicate in Petri dishes containing 25 mL of malt extract agar (ISP2 medium (g L^−1^) [40,41]: yeast extract, 4 g; malt extract, 10 g; dextrose, 4 g; agar, 20 g) of pH 7.2 ± 0.2 sterilized at 121 °C and 0.1 kg_f_ cm^−2^ for 15 min (Hirayama H1CLAVE HV 50, Amerex Instruments, Lafayette, CA, USA). The plates were incubated at 30 °C until colonies with a hard texture and formation of mycelial branches were observed (approx. 7–15 days) (Digital Incubator Felisa FE-132D, Fabricantes Feligneo, Mexico). The microbial culture was used to prepare spore stocks and inoculums for submerged cultures.

#### 2.1.1. Preinoculum

Cell growth on one side of a 0.5 cm^2^ agar cube was inoculated in triplicate in 125 mL Erlenmeyer flasks with 50 mL of ISP2 culture medium prepared as mentioned before but without agar. The flasks were incubated at 30 °C and 200 rpm for 72 h (Thermo Forma 420 Incubator Shaker, Forma Scientific Inc., Marjetta, OH, USA), or until abundant cell growth was observed. The purity of the culture was verified by microscopic observation (Olympus, Model CH30RF100, Tokyo, Japan).

#### 2.1.2. Spore Stock

The microbial growth in Petri dishes was resuspended with sterilized distilled water, collected, and filtered (Millipore membrane 0.22 μm) for the spore preservation in 1 mL Eppendorf tubes. The purity of the spore stock was verified by microscopic observation and reseeding on malt extract agar. The spore vials were stored at 4 °C.

### 2.2. Biomass Production

#### 2.2.1. Vegetative Cell Production

A pre-inoculum (mycelium fragments, vegetative cells, and filaments) of 10 mL and 2 × 10^5^ ± 1.2 × 10^4^ CFU mL^−1^ (CFU: colony-forming unit) was poured into 500 mL of culture medium (YGM) based on yeast extract, glucose, and mineral salts (composition in g L^−1^: yeast extract, 6 g; glucose, 10 g; Na_2_HPO_4_, 5.3 g; KH_2_PO_4_, 1.98 g; MgSO_4_ 7H_2_O, 0.2 g; NaCl, 0.2 g; CaCl_2_ 2H_2_O, 0.05 g, and 1 mL of trace element solution with the following composition in grams per 100 mL of distilled water: CuSO_4_ 5H_2_O, 0.64 g; FeSO_4_, 0.11 g; MnCl_2_ 4H_2_O, 0.79 g; ZnSO_4_ 7H_2_O, 0.15 g), at a pH of 7.1–7.2, and contained in a 1 L flask. The culture, prepared in triplicate, was incubated at 30 °C and 200 rpm (approx. 4 days). Microbial culture was drained at the end of the exponential growth phase and the biomass was separated by centrifugation (Sorvall 6000, Thermo Scientific, Waltham, Massachusetts USA) at 9000 rpm and 4 °C for 15 min. The culture medium absorbed in the biomass was removed with sterilized distilled water by shaking (Vortex agitator model Vortex 2, IKA Works Inc., Staufen, Germany) and centrifugation. The biomass was resuspended by shaking in 50 mL of water and poured into vials for storage at 4 °C until immobilization in calcium alginate capsules. The microbial culture purity was verified by serial dilution of a biomass sample (1 mL) and seeding on malt extract agar (Petri dish). The microbial count determined in triplicate by serial dilution was 2.5 × 10^5^ ± 4.73 × 10^4^ CFU mL^−1^. Microbial cultures with a visible concentration of spores (4 × 10^5^ spores mL^−1^) were discarded.

#### 2.2.2. Spore Production

Aliquots of 100 µL were taken from the pre-inoculum (Section 2.1.1) and seeded into 10 Petri dishes prepared with Difco^®^ Potato Dextrose Agar (PDA). The Petri dishes were incubated at 30 °C for 7 days or until evident sporulation of microbial growth was observed.

The biomass was resuspended by shaking in 15 mL of sterile distilled water. The spores in the supernatant were recovered by filtration (Millipore membrane 0.22 μm) and poured into vials of 50 mL for conservation at 4 °C until immobilization in calcium alginate capsules. The spore count in the vials (9.25 × 10^5^ ± 3.5 × 10^4^ spores mL^−1^) was determined by a serial dilution from a 1 mL biomass sample and counted under the microscope in a Neubauer chamber (Bright Line, Optik Labor, Görlitz, Germany), and CFU counted on PDA medium. The spore suspension was homogenized by sonication (10 min) to ensure reproducibility of the microbial count.

### 2.3. Immobilization of Streptomyces CDBB1232 on Calcium Alginate Capsules

Vegetative cells and spores of *Streptomyces* CDBB1232 were separately immobilized in calcium alginate according to the methodology reported by Mancera-López et al. (2019) [42]. The concentrated biomass (mycelium (vegetative cells) or spores) produced as described in Section 2.2, was mixed under stirring conditions with 200 mL of a sterilized solution of 1.5% sodium alginate and 200 mL of YGM medium; NMR analysis of the sodium alginate, which was acquired from CIVEQ, Mexico City, Mexico, showed a predominance of mannuronic acid relative to guluronic acid as shown in Figure 1. The analysis was performed on a Jeol ECA-500 spectrometer at 500.159 MHz in D_2_O solution. Spectral at 6510.41 Hz, acquisition time 5.03 s, 32,768 points, 256 scans, and a recycle delay of 1 s. The assignments were in agreement with Behfar et al. (2022) [43]. The *Streptomyces* CDBB1232 suspension was extruded dropwise using a peristaltic pump (PCS^®^, model PP101-1, Temecula, CA, USA) into a sterilized 0.2 M calcium chloride solution maintained with gentle agitation (Figure 2); droplet size (2–3 mm in diameter) was controlled with a hypodermic needle (31 G × 8 mm and 20 G × 9.5 mm). After one hour in the calcium chloride solution (25 ± 2 °C and gentle shaking), the calcium alginate capsules were separated by filtration (medium-pore filter) and washed with sterilized distilled water before their storage at 4 °C until later use. The morphological analysis of the fresh capsules was performed using a stereo microscope (Leica L2 MZ6 stereo microscope. Wetzlar, Germany). Before and after immobilization of the biomass, the spore count and the colony-forming units (CFUs) in the case of the mycelium were determined. The immobilized biomass concentration was quantified after solubilizing the calcium alginate matrix with 1% sodium citrate (6 capsules in 1 mL of sodium citrate). The microbial count was performed with serial dilution and CFU counting on PDA as previously described. All the experiments were carried out in duplicate, with sterile materials, and in a sanitized environment.

### 2.4. Dehydration Process of Encapsulated Biomass

The immobilized biomass was dried in a 3.5 L rotary drum (12 cm internal diameter × 30 cm length) as presented in Figure 3. The drum, previously sterilized with UV irradiation of 250 nm for 15 min, was loaded with 125 ± 6 g of biomass immobilized on the calcium alginate capsules for drying under controlled conditions for airflow (2–10 L min^−1^) and rotation speed (3.5 rpm); the air was fed at room temperature (24.5 ± 2.5 °C). The drying process was carried out by convection feeding a sterilized air flow to the reactor through a sterilized porous stone (pore size 40–100 µm). A humidity of 20 ± 2% in the airstream at the outlet of the rotating drum was the criterion to stop the drying process. Relative humidity (% RH) and air temperature were recorded every 4 h using a hygro-thermo-anemometer (Hygro Thermo-Anemometer Extech Model AM-4205, Nashua, New Hampshire, USA) placed in an air chamber at the inlet and outlet of the drum. The most suitable airflow to dehydrate the immobilized biomass was determined from data of the strain viability as a function of the airflow (2, 4, 6, 8, 10 L min^−1^). The dehydrated capsules were placed in Petri dishes for stabilization at 30 °C for 24 h. The water mass removed at the end of the dehydrated process was determined by the difference in weights.

The viability of the encapsulated and dehydrated biomass was determined by comparing the biomass counts (mycelium or spores) as CFU mL^−1^, before and after the dehydration process. Following this, 0.1 g samples of dried capsules were hydrated in 1 mL distilled sterilized water and the biomass subsequently released from the calcium alginate by its solubilization with a 1% sodium citrate solution (9 mL); the biomass release was promoted with vortexing for 10 min. The calcium alginate residues were separated by centrifugation at 5000 rpm for 10 min, and the supernatant was used for the analysis of microbial count by serial dilution. Then, 100 μL samples of each dilution were seeded in Petri dishes with PDA medium for microbial growth at 30 °C for 48 h. Biomass viability was calculated as follows:(1)$$Biomass\;viability\left(\%\right)=\frac{\mathrm{viable}\;\mathrm{biomass}\;\mathrm{count}\;\mathrm{after}\;\mathrm{capsule}\;\mathrm{dehydration}}{\mathrm{viable}\;\mathrm{biomass}\;\mathrm{count}\;\mathrm{before}\;\mathrm{capsule}\;\mathrm{dehydration}}\times 100$$

The morphological analysis of the dehydrated capsules was performed using scanning electron microscopy (SEM). A sample of capsules was placed in copper support where they were adhered with a double-sided graphite tape and covered with a gold plating. The samples were directly observed in the SEM JEOL model JSM-6510LV under 25 KV (JEOL, Peabody, MA, USA). The dehydrated capsule diameter was determined by image analysis (ImageJ 1.46 r software (public domain), Tiago Ferreira and Wayne Rasband).

### 2.5. Additional Protectors

Trehalose, kaolin, and gum arabic are materials that have been used in combination with other polymers to protect biological agents from dehydration treatments [24,34,37,44,45,46]. In the present study, these were used separately in different stages of the immobilization process of the *Streptomyces* CDBB1232 mycelium (Appendix A). Trehalose was added at a concentration of 10 g L^−1^ in the submerged culture of *Streptomyces* CDBB1232. Kaolin and trehalose were both added at a concentration of 10 g L^−1^ to the 1.5% sodium alginate solution. In these three cases, the encapsulation process was carried out according to the methodology described in Section 2.3. The gum arabic was impregnated in the calcium alginate capsules immediately after its separation by filtration from the calcium chloride solution and washing with distilled water. This protector was applied in the solution (4%) for 30 min. The capsules were separated by filtration and washed with 0.1% saline solution previously sterilized.

### 2.6. Control Experiments

Two control experiments were implemented. The first involved dehydration of the calcium alginate capsules (empty and with spores) with airflow at room temperature. The operating conditions of the rotating drum were the same as those applied to the dehydration of the immobilized mycelium. The second control comprised the dehydration of the alginate capsules in an oven and controlled temperature at 30 °C. In the heat dehydration process, the mass of the capsules was the same as that of the capsules dehydrated in the rotary drum. Dehydration time, moisture removed, and strain viability, if applicable, were recorded for each batch of dehydrated material. All experiments were carried out in duplicate with sterile materials and under sterile conditions.

### 2.7. Analysis of the Drying Process

The water mass removed from the calcium alginate capsules was determined, as well as the diffusion coefficient of water entrained by the air flow fed to the rotating drum. The humidity mass was determined with the temperature and relative humidity data recorded at the inlet and outlet of the rotating drum [47]. The diffusion coefficient was determined from the equation proposed to determine the flow of water through air [48].

### 2.8. Statistical Analysis

The experiments were carried out in duplicate, and the results reported are the mean ± standard error (S.E). Statistical analysis was performed with Minitab 18 version. The Tukey test determined the significant differences between the means (*p* < 0.05).

## 3. Results

### 3.1. Biomass Production

*Streptomyces* CDBB1232 was propagated in liquid culture with an average yield of 4 ± 1 g dry biomass per liter over four days. The length and thickness of the mycelium changed during the first two days of the culture toward the formation of well-developed heterogeneous spongy granules. A dense network of branched mycelia and active filament growth, with no evidence of spore production, was observed under the microscope (Figure 4A). In nutrient agar, the mycelium formed characteristic colonies of the *Streptomyces* genus in three days, and the microbial count was 2.5 × 10^5^ ± 4.7 × 10^4^ CFU mL^−1^. Colonies were dry and white with an irregular surface of powdery texture and filamentous edges (Figure 4B). The average diameter of the colonies was 5 mm and a beige pigmentation spread under these was observed.

In liquid cultures of more than four days, the production of spores was observed. The spores were seeded in nutrient agar medium; they did not develop into mycelia but developed into small mucous colonies including exclusively more spores (Figure 4C).

### 3.2. Immobilization of Streptomyces CDBB1232 in a Calcium Alginate Matrix

Immobilization was carried out with the biomass (pellets or spores) suspended in a sodium alginate solution adjusted to a concentration of 7.6 × 10^6^ ± 5.7 × 10^5^ CFU mL^−1^. The weight of the capsules produced was 125.4 ± 6 g with a diameter of 3.3 ± 0.7 mm (Figure 5). The average microbial counts of the mycelium and spores, after release from their respective calcium alginate matrixes, were 6.69 × 10^5^ ± 5.81 × 10^4^ CFU g^−1^ and 7.27 × 10^5^ ± 2.11 × 10^4^ CFU g^−1^, respectively. Compared to the microbial count before immobilization, the biomass lost its viability in about one order of magnitude.

### 3.3. Dehydration of Immobilized Biomass in Alginate Capsules

The dehydration process was carried out with the capsules produced in each batch, 125.4 ± 6 g. The humidity loss profile of the immobilized spores was linear, while that of the mycelium presented a slight curvature. The ANOVA of the humidity loss profiles of the two types of biomass immobilized by the effect of the airflow variation showed significant differences for an α < 0.05 (mycelium: F = 970.287, df = 13, *p* < 0.001; spores: F = 765.555, df = 13, *p* < 0.001). The higher the airflow rate, the faster the humidity removal rate and the shorter the time required to reach the residual humidity of 20%, which was the minimum humidity required to maintain the viability of the strain (Figure 6). The type of biomass (cells/spores) did not generate significant differences in the humidity removal profiles (F = 0.198, df = 1, *p* = 0.658). However, some differences were observed at 2 and 10 L min^−1^ air flows, but only during the first four hours and regarding the immobilized mycelium.

Two-way ANOVA (Tukey’s test for an α < 0.05) (F (mycelium) = 970.287, df = 13, *p* < 0.001; F (spores) = 765.555, df = 13, *p* < 0.001; F (mycelium, spores) = 0.198, df = 1, *p* = 0.658).

The rate of humidity removal during the drying process of the encapsulated biomass is presented in Figure 7. This was proportional to the airflow fed to the rotating drum up to an airflow of 8 L min^−1^; between airflows of 8 and 10 L min^−1^ no significant differences were observed. The rate of moisture removal decreased over time and depended on the residual moisture in the encapsulated biomass. Under airflows of 6 L min^−1^ and higher, the lowest rate of humidity loss was above 1 g h^−1^, while under airflows of 2 and 4 L min^−1^, the rate was 0.35 g h^−1^. Moisture loss at a rate higher than 4 g h^−1^ (Figure 8) led to the formation of cracks on the capsule surface and consequently to the mycelium exposure as observed in Figure 8B. Under a humidity loss rate lower than 4 g h^−1^, no significant damage was observed except in capsules prepared without biomass (Figure 9). The type of encapsulated biomass did not affect the rate of humidity removal. In comparison with the initial humidity, the dehydration process of the immobilized biomass allowed 94 ± 1% of water removed. Neither airflow nor moisture removal with heat affected the average diameter of the dehydrated capsules (having biomass) to a minimum moisture of 20% (F = 1.83; df = 2; *p* = 0.177; Tukey’s test for an α < 0.5). The average capsule diameter was 1.06 ± 0.14 mm. The diameter of the capsules without biomass was lower by 21%.

The diffusion coefficient of the water removed from the encapsulated mycelium with different air flows only showed significant differences in relation to the time under which the capsules were exposed (F = 59.08; df = 13; *p* < 0.001), as shown in Figure 10. The profiles of the diffusion coefficient were consistent with the phenomenon analyzed: the lower the mass of residual water in the capsules, the lower the amount of water to be eliminated through the surface of the capsules per unit of time.

### 3.4. Viability of Encapsulated and Dehydrated Streptomyces CDBB1232

The viability loss of the immobilized biomass was analyzed as a function of the airflow fed to the rotating drum and biomass type. Of these two parameters, the airflow impacted the mycelial viability to a greater extent than in the spores, as seen in Figure 11. The loss of viability was observed in PDA cultures of the immobilized mycelium dehydrated with air flows from 6 L min^−1^. Although the encapsulated mycelium dehydrated at 6 L min^−1^ grew in beige colonies with a creamy appearance, these colonies only contained spores that did not form germ tubes during 15 days of incubation. The immobilized mycelium exposed to air flows of 8 and 10 L min^−1^ did not propagate, possibly because of the fragmented and damaged mycelium observed under the microscope.

The encapsulated and dehydrated spores were more resistant to air drying than the mycelium, with the maximum viability loss of 60 ± 6% even under an airflow rate of 10 L min^−1^. The colonial morphology observed in the cultures presented in Figure 12 allowed for verifying the conservation of the *Streptomyces* spore viability and consequently their resistance to the encapsulation and dehydration process to which they were subjected.

### 3.5. Heat Dehydration of Immobilized Biomass

The immobilized mycelium and spores of *Streptomyces* CDBB1232 were dehydrated at 30 °C for 120 ± 6 h until reaching a residual humidity of 20%, and the dehydrated capsule size was similar (1.06 ± 0.14 mm) to that obtained with air at room temperature (Figure 13). This dehydration method also caused the fracture of the alginate capsule surface (Figure 13B), and the viability losses of the mycelium and spores were determined in percentages of 23% and 10%, respectively; in nutrient agar, both the mycelium and spores propagated until the spore formation. The time required to dehydrate calcium alginate capsules free of any active component only decreased by five hours compared to the time required to dehydrate capsules with the immobilized biomass. The same time difference was observed when the calcium alginate capsules, with and without the active agent, were air-dried.

### 3.6. Effect of Protectors in the Dehydration with Air of Encapsulated Streptomyces CDBB1232 Mycelium

The inclusion of kaolin and gum arabic as additional protectors in the YGM medium in the calcium alginate capsules did not result in significant differences in the viability of *Streptomyces,* unlike when only the YGM medium was used (Table 1). However, of these two additives, only gum arabic successfully protected the encapsulate from the air drying, after which no changes in the viability of the strain were observed; with the kaolin the strain lost its viability by 96%. The protective activity of trehalose was highlighted when it was included as a substrate in the *Streptomyces* culture but only in the encapsulation process because this effect did not extend to the drying process, where the loss of viability was 90%. As an additive to the alginate matrix, trehalose allowed an increase in the viability of *Streptomyces* by 25%; however, in the air-drying process, trehalose did not improve the viability percentages obtained with the YGM medium. SEM analysis of the dehydrated capsules prepared with trehalose and kaolin in sodium alginate showed the presence of crystals of both components on the capsule surface (Figure 14A,C, X5000 5 μm). The formation of cracks on the surface of the dehydrated capsules was attributed to the crystallization of both these additives.

According to the results obtained, only the gum arabic generated satisfactory results with negligible viability losses during the dehydration treatment. The preservation of viability was promoted by the pore sealing of the calcium alginate matrix. The SEM analysis of these capsules suggested that the exposed mycelium was possibly wrapped by the gum arabic, preventing its exposure to the friction of the air fed to the rotating drum (Figure 14D, X5000 5 μm). The treatment of the capsules with gum arabic protected the viability of the encapsulated biological material by 87%.

## 4. Discussion

*Streptomyces* is a strain that requires careful selection of the culture medium as well as environmental conditions that favor its propagation and production of the desired metabolites [49,50,51]. The YGM (yeast extract, glucose, and mineral salts) medium is designed to produce biomass, and in the present work, it generated a yield of 4 ± 1 g L^−1^ of *Streptomyces*, which was consistent with what was reported in the literature. Sabaratnam and Traquair (2002) [44], for example, reported a yield of 4.8 g L^−1^ in three days of *Streptomyces* incubation in a YGM medium. Maldonado et al. (2010) [52] propagated *Streptomyces* RO3 in both YGM media and in a medium based on starch and casein. The yields reported by these authors were 2.15 and 0.477 g L^−1^, respectively, in 3–5 days of incubation. In liquid culture, *Streptomyces* grew in the form of spongy and heterogeneously sized pellets as reported in the literature [51,53]. Its propagation in dispersed cells is possible [44]; however, the formation of pellets is considered a self-protection mechanism that preserves its viability against shear phenomena between the mycelium and the moving liquid medium [51].

The slow propagation in submerged cultures of *Streptomyces* has been attributed to the high viscosity of the culture medium, which limits the transfer of nutrients and oxygen to the cells located in the pellet center [51,54,55]. This low availability of nutrients is a factor that triggers the production of spores, and even these can be non-viable when the strain is grown in submerged culture [56,57]. In the present study, the production of non-viable spores was observed in cultures of more than four days. This observation was the criterion to stop the *Streptomyces* culture and proceed to its immobilization in calcium alginate. In a study of the spores’ viability produced by *Streptomyces antibioticus* in liquid medium, van Dissel et al. (2014) [58] reported viability losses of 20% in spores produced 5 h after their induction.

The *Streptomyces* mycelium was encapsulated in calcium alginate with YGM medium as a protective agent. The YGM medium is a low-cost protective agent compared to other additives such as trehalose [34]. The drawback of culture media is their formulation, which may contain anti-gelling substances such as Mg^+2^ [38,59]. In the present study, the YGM medium included MgSO_4_ 7H_2_O; however, the alginate gels formed presented a firm consistency, which allowed us to assume that its anti-gelling effect could be of low magnitude. The elements of interest in culture media are the sugars, and their function in a microbial formulate is to protect the cell membrane from microorganisms subjected to a drying treatment. In this work, *Streptomyces* spores were resistant to the combined encapsulation and dehydration process, and their good state allowed the strain to complete their life cycle. The mycelium, on the other hand, lost its viability mostly during the humidity removal under air flows higher than 4 L min^−1^.

Some factors that could have contributed to the loss of mycelium viability during the implemented encapsulation process are the following:Cellular shear damage. The high concentration of the biomass and the high viscosity of the alginate solution demanded vigorous stirring to produce a homogeneous suspension of the heterogeneously sized pellets for the capsule production by extrusion. The high shear force, in addition to causing cell stress, can cause cell damage. The loss of viability due to shearing of cellular material has been already reported by Sabaratnam and Traquiar (2002) [44] and Tamreihao et al. (2016) [60].Cellular stress due to mechanical dispersion of pellets. In liquid culture, *Streptomyces* grows as pellets. The variable diameter of the pellets demanded their mechanical dispersion in order to facilitate their flow through the extrusion device. Mechanical dispersion is a reason for cellular stress and consequently of viability loss [51].Stress due to rapid dehydration of immobilized biomass. A high rate of moisture removal was observed during the first 8 h after the dehydration process began and under feeding air flows of 6 L min^−1^ and higher. Under these air flows, water diffusion coefficients between 8 × 10^−5^ and 14 × 10^−5^ m^2^ s^−1^ were determined, which were higher than those reported in the literature for fruit dehydration [61]. The viability loss of the immobilized mycelium was registered after the capsule dehydration under air flows equal to or greater than 6 L min^−1^. It has been suggested that accelerated humidity loss in polymeric materials such as alginate causes the formation of a crust that shrinks rapidly toward the capsule core and subsequently slows the rate of humidity removal [62]. The rapid contraction of the capsule surface can cause cell damage and cell death because of lack of oxygen [63]. In the present work, under an air flow of 10 L min^−1^, the deceleration of the humidity removal process by contraction of the alginate crust was detected by the change in slope in the humidity loss profile, which decreased from 12 to 3.6 g h^−1^. The lower loss of the dehydrated mycelium viability under an air flow of 2 L min^−1^ can be explained by the long strain adaptation process that lasted approximately 10 h before the water loss was recorded.

The higher the temperature, the shorter the dehydration time; however, Brun-Graeppi et al. (2011) [64] reported that the accelerated humidity removal with heat leads to the formation of cracks in the polymeric membrane and consequently to the loss of biological activity of the cell biomass. This phenomenon was observed in the control experiments implemented in the present work, even when the dehydration with heat was conducted only about 5 °C above room temperature.

Transfer of nutrients and oxygen to the capsule core. Biological material immobilized in alginate can give rise to low percentages of viability considering that the availability of nutrients to the nucleus of the capsule will depend on the cell concentration, the capsule diameter, and the capsule porosity. In the present work, the micellar viability was determined by release of the biological material, as well as by dispersion with grinding of the fresh and dehydrated capsules. Both methods yielded similar viability percentages, although with microbial counts that differed by one power; the highest microbial count was determined in the samples of biological material dispersed by grinding.

McLoughlin (1994) and Rathore et al. (2013) [65,66] reported poor viability in polymeric capsules greater than 1 mm in diameter. These authors attributed the low viability of the strain to the low availability of nutrients, the accumulation of toxic metabolites, and cell death because of lack of oxygen [67,68,69]. The average diameter of the dehydrated capsules obtained in the present work was 1 mm. The seeding of these in nutrient agar and PDA and the observation of a homogeneous production of cellular material on the capsule surface allowed verifying the transfer of nutrients through the alginate wall.

Biomass instability. Manteca et al. (2008) and López-García et al. (2014) [70,71] reported that culture instability can lead to rapid aging of the mycelium and subsequent disintegration of dead hyphae. In the present work, biological material resistant to the encapsulation process was obtained by immobilizing cultured cellular material obtained at the end of its exponential growth phase. Vriezen et al. (2006) and Schoebitz et al. (2012) [24,72] reported that the mycelium harvested in this phase is stable and more resistant to stress caused during handling.

Water evaporation, lyophilization, and water dragging with air are the methods mostly applied to the dehydration of biological material, whose performance is determined by the operating variables’ temperature and air flow [14,17,19,73]. Of these methods, the removal of moisture with heat is efficient; however, high evaporation temperatures are a reason for the viability loss of the biological material [74]. Schoebitz et al. (2012) [24] reported a viability loss of two orders of magnitude in the *Azospirillum brasilense* strain (rhizobacteria) immobilized in calcium alginate when dehydrated at 35 °C for 24 h. Costa et al. (2002) [75] reported a loss of viability between two and five orders of magnitude of dehydrated material encapsulated in alginate of *Pantoea aglomerans,* which was dehydrated by spray at 140–170 °C and air flow of 500 mL h^−1^. Regarding the moisture removal with air, moisture is usually removed with air flows lower than 2 L min^−1^, and mostly at a temperature higher than room temperature [31]. The viability loss of the biomass dehydrated by this technology was associated with the characteristics of the same biomass [17]. That is, in the case of the mycelium, the loss of viability can be very high, while in the case of spores, practically negligible [27,60]. In the present work, the drying process conducted with air flows in the range of 2–10 L min^−1^ allowed us to identify the flow of 6 L min^−1^ as critical to carrying out the dehydration of the *Streptomyces* mycelium immobilized in alginate and using YGM medium as protector. Under an air flow of 4 L min^−1^, the loss of viability was 84%; however, this decreased to 13% by layering gum arabic over the surface of the alginate capsules.

Despite the reported instability of calcium alginate as an encapsulating support for biological material [38], it continues to be used preferentially for its biocompatibility and safety properties with biological material, as well as the possibility of improving the properties of the alginate either by mixing with other polymers or by including various additives in the alginate matrix [35,36]. In the present study, which was focused on the development of an encapsulated and dehydrated formulation of the Streptomyces CDBB1232 mycelium, of the different additives integrated into the calcium alginate matrix (trehalose, kaolin, and gum arabic), only the gum arabic protected the mycelium from the air-drying process. The inclusion of this polymer in the formulation allowed preservation of the viability of the Streptomyces mycelium by 87%. Viability percentages of 94% in Streptomyces fulvissimus Uts22 encapsulated in a calcium alginate support mixed with gum arabic and nanoparticles (SIO2 and TIO2) was reported by Saberi-Riseh et al. (2022) [4]. According to these authors, who applied a layer-by-layer encapsulation technique, the interaction between calcium alginate and gum arabic is of an electrostatic type. Similar encapsulation efficiencies using whey protein and gum arabic were reported by Sharifi et al. (2021) [76] in the encapsulation of Lactobacilli by the complex coacervation method. According to the encapsulation efficiencies reported by these authors and those obtained in the present work, the gum arabic can form a protective material in encapsulation processes, regardless of the process applied. In an encapsulation formulation, the sugars have the function of protecting the vegetative cells against osmotic pressure and desiccation during the capsule-drying step [17,34,77]. This type of protection occurs because of the potential of sugars to be absorbed and aggregate in the cytoplasm [14]. An example is Pseudomonas fluorescens immobilized on a solid support; after air flow dehydration, the bacterium lost its viability by three orders of magnitude [78]. using fructose and trehalose as protective substances, the percentage of the surviving dehydrated bacteria increased [72]. The viability loss of bacteria exposed to a drying process without the use of protective substances has also been reported by Ghandi et al. (2013) [79]. Glucose and trehalose are substances commonly used as protectors in encapsulated microbial formulations. However, although trehalose has been reported to be the most effective [19,77,80], this sugar is not applicable for all strains, especially if trehalose is metabolized by the microorganism rather than accumulating it [77]. In the particular case of Streptomyces, there are reports on its ability to produce and metabolize trehalose [81]. In the present study, the inclusion of trehalose in the alginate capsules, both as a growth nutrient and as a component of the alginate matrix, only showed protective effects in the encapsulation operation.

Kaolin is one of the most used inert supports in the immobilization of microbial strains and an additive in the encapsulation with alginates of microbial strains [13,34]. In microbial formulations, kaolin plays the role of a protective wall against irradiation phenomena and heat treatments of the biomass. In Streptomyces griseus formulations, Zacky and Ting (2015) [13] reported a better performance of kaolin compared to kaolin as an alginate additive. The same results were reported by Sabaratnam and Traquair (2002) [44], but with *Streptomyces* sp. Di-944 immobilized in talcum powder and granules of wheat flour mixed with starch. The results obtained in this work are consistent with the results reported by these authors. However, it is important to consider the preparation process of the formulations. In the papers consulted, the biomass was immobilized in the kaolin prior to encapsulation. In the present work, the large size of the pellets formed by the Streptomyces mycelium did not allow for this prior immobilization, which could have prevented the kaolin from playing its protective role together with the calcium alginate.

## 5. Conclusions

Encapsulation and dehydration are two of the technologies applied in the conservation of microbial biological control agents. The present work was focused on the development of a viable and economical methodology for the encapsulated and dehydrated formulation of *Streptomyces* CDBB1232. The encapsulation was carried out by extrusion with sodium alginate solubilized in a YGM culture medium (yeast, glucose, and mineral salts) and dehydration in a rotary drum fed with an air flow of 4 L min^−1^. The common formation of cracks during the dehydration process of alginate capsules was prevented by adding gum arabic to the surface of fresh capsules. The implemented encapsulation method allowed us to produce viable formulations (87%) with an average particle diameter of 1 mm. The microbial count of the formulation per unit mass, as well as the characteristics of the formulation and the low production costs, can be of interest for its application at the field level.

## Figures and Tables

**Figure 1 polymers-15-00207-f001:**
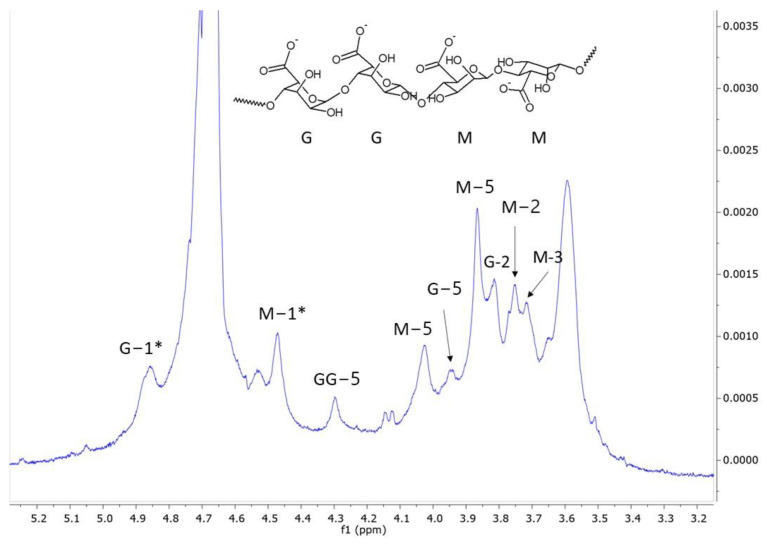
^1^H nuclear magnetic resonance spectrum with partial presaturation of residual H_2_O signal of sodium alginate (acquired from CIVEQ, Mexico City, Mexico) that was used to immobilize the *Streptomyces* strain (spores and mycelium). G−1* and M−1* indicate the anomeric hydrogen of glucuronic acid and mannuronic acid respectively.

**Figure 2 polymers-15-00207-f002:**
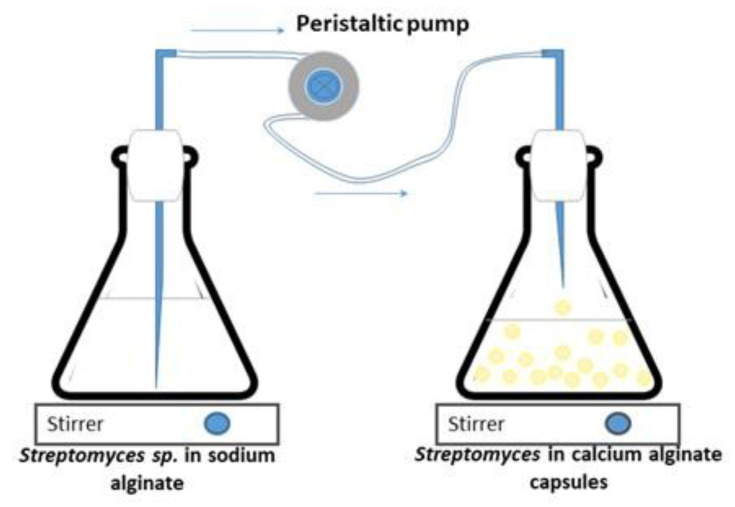
Immobilization of *Streptomyces* CDBB1232 in calcium alginate capsules.

**Figure 3 polymers-15-00207-f003:**
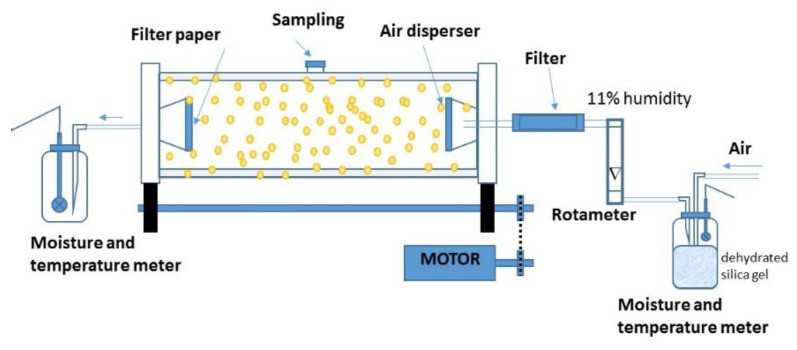
Rotating drum dryer.

**Figure 4 polymers-15-00207-f004:**
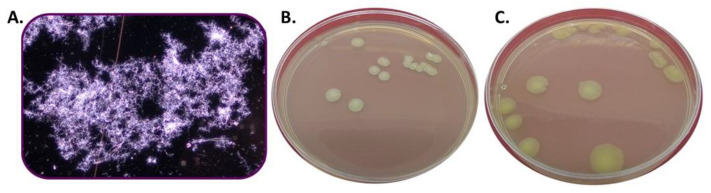
*Streptomyces CDBB1232*. (**A**) Viable mycelium grown in YGM liquid culture (100× microscope view). (**B**) Colonial morphology of a viable culture; (**C**) Colonial morphology of a non-viable culture.

**Figure 5 polymers-15-00207-f005:**
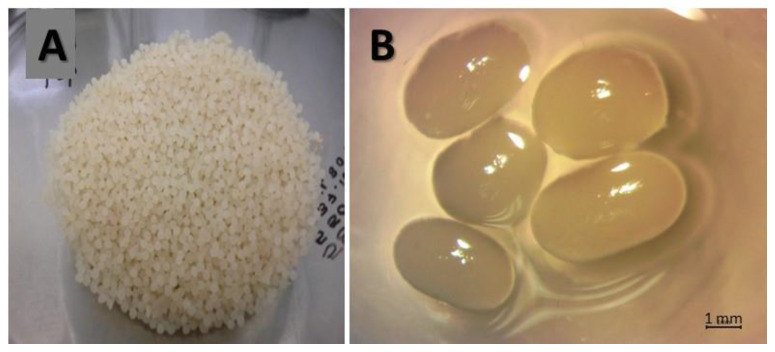
Immobilized biomass in calcium alginate capsules. (**A**). Filtered alginate capsules, (**B**). Size and form of fresh capsules.

**Figure 6 polymers-15-00207-f006:**
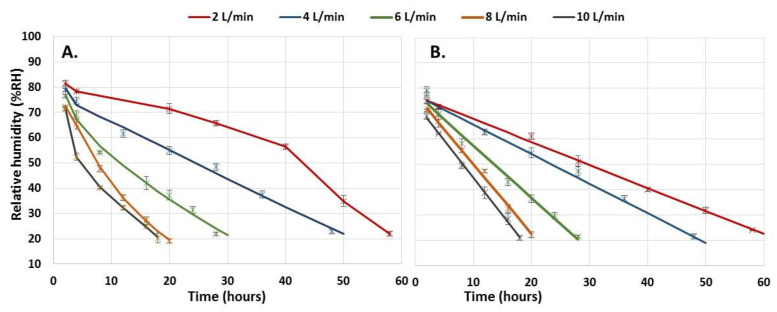
Dehydration profiles of immobilized biomass in calcium alginate capsules. (**A**)**.** Mycelium, (**B**)**.** Spores. Error bars represent the standard deviation of two determinations.

**Figure 7 polymers-15-00207-f007:**
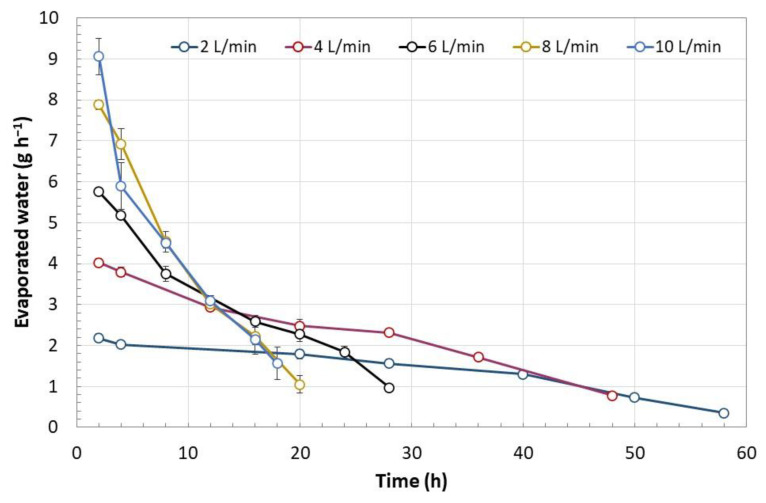
Drying speed of the biomass immobilized in calcium alginate as a function of the time required to reach a residual humidity of 20%. Error bars represent the standard deviation of two determinations. Two-way ANOVA (Tukey’s test for an α < 0.5) (F = 1.83; df = 2; *p* = 0.177).

**Figure 8 polymers-15-00207-f008:**
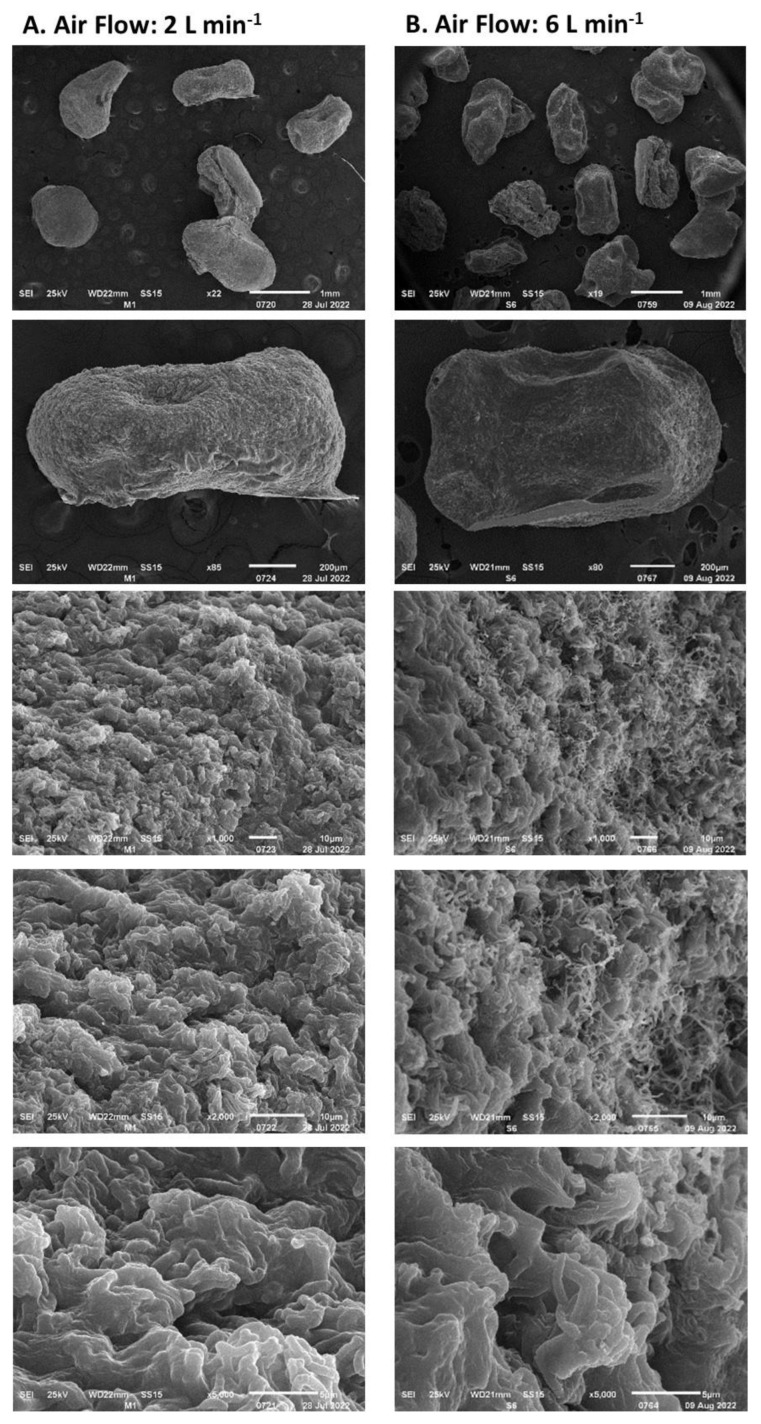
The *Streptomyces CDBB1232* mycelium encapsulated and dehydrated with airflow: (**A**). 2 L min^−1^, (**B**)**.** 6 L min^−1^, at room temperature.

**Figure 9 polymers-15-00207-f009:**
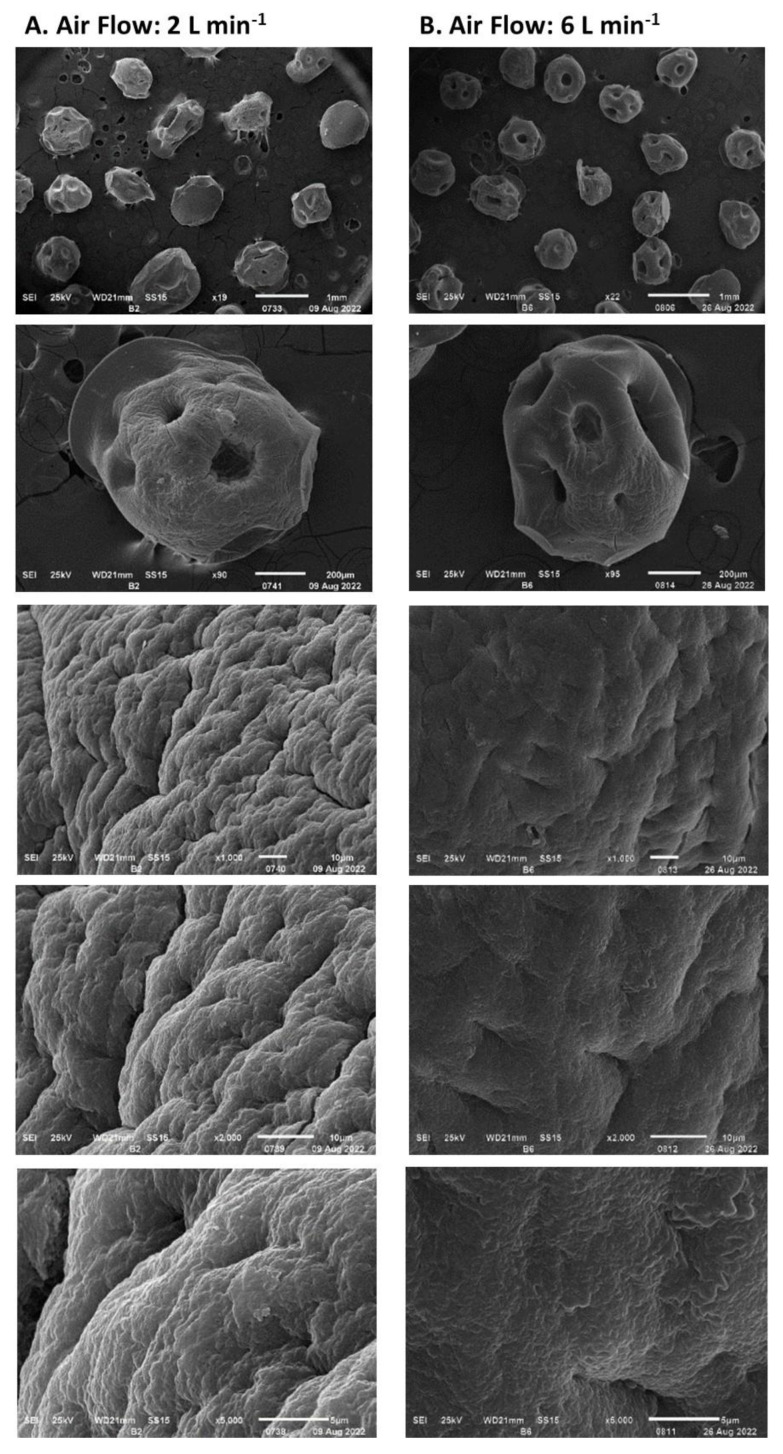
Calcium alginate capsules dehydrated with airflow ((**A**). 2 L min^−1^, (**B**). 6 L min^−1^) at room temperature.

**Figure 10 polymers-15-00207-f010:**
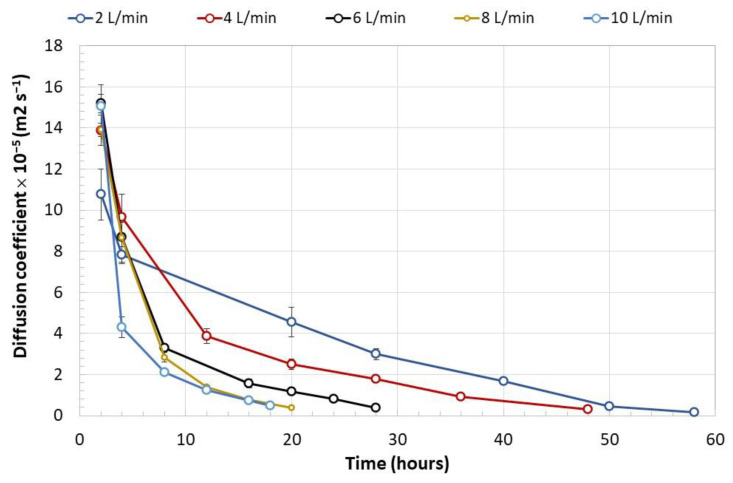
Diffusion coefficient (m^2^ s^−1^) of the water withdrawn from the immobilized *Streptomyces* mycelium under different air flows and over time. Error bars represent the standard deviation of two determinations. Two-way ANOVA (Tukey’s test for an α < 0.05). F (time) = 59.08; df = 13; *p* < 0.001, F (airflow) = 2.47; df = 4; *p* = 0.057.

**Figure 11 polymers-15-00207-f011:**
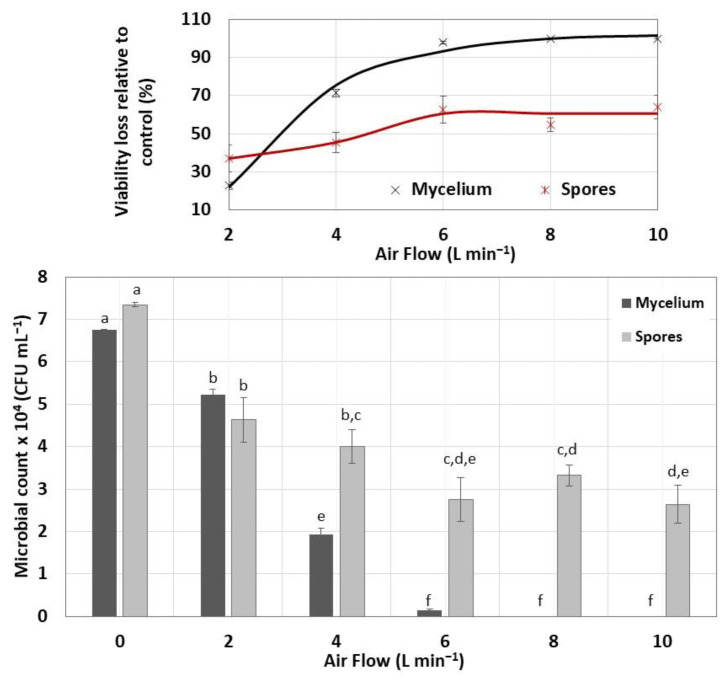
Effect of the airflow velocity on the viability of the immobilized biomass dehydrated in a rotary drum-type dryer. The mixed ANOVA indicated differences in the viability of *Streptomyces* CDBB1232 caused by the effect of the airflow and biomass type (significant effect of the airflow: F = 196.57, df = 5, *p* < 0.001; mixed effect (airflow and biomass): F = 39.7, df = 5, *p* < 0.001). Bars labeled with the same letter do not significantly differ (Tukey’s test for an α = 0.05).

**Figure 12 polymers-15-00207-f012:**
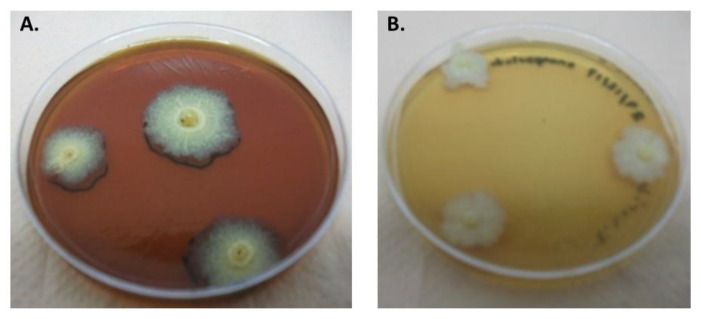
*Streptomyces* CDBB1232 spore cultures after being subjected to the processes of (**A**) immobilization in calcium alginate and dehydration with an airflow of 4 L min^−1^, (**B**) immobilized biomass dehydrated at 30 °C for 6 days.

**Figure 13 polymers-15-00207-f013:**
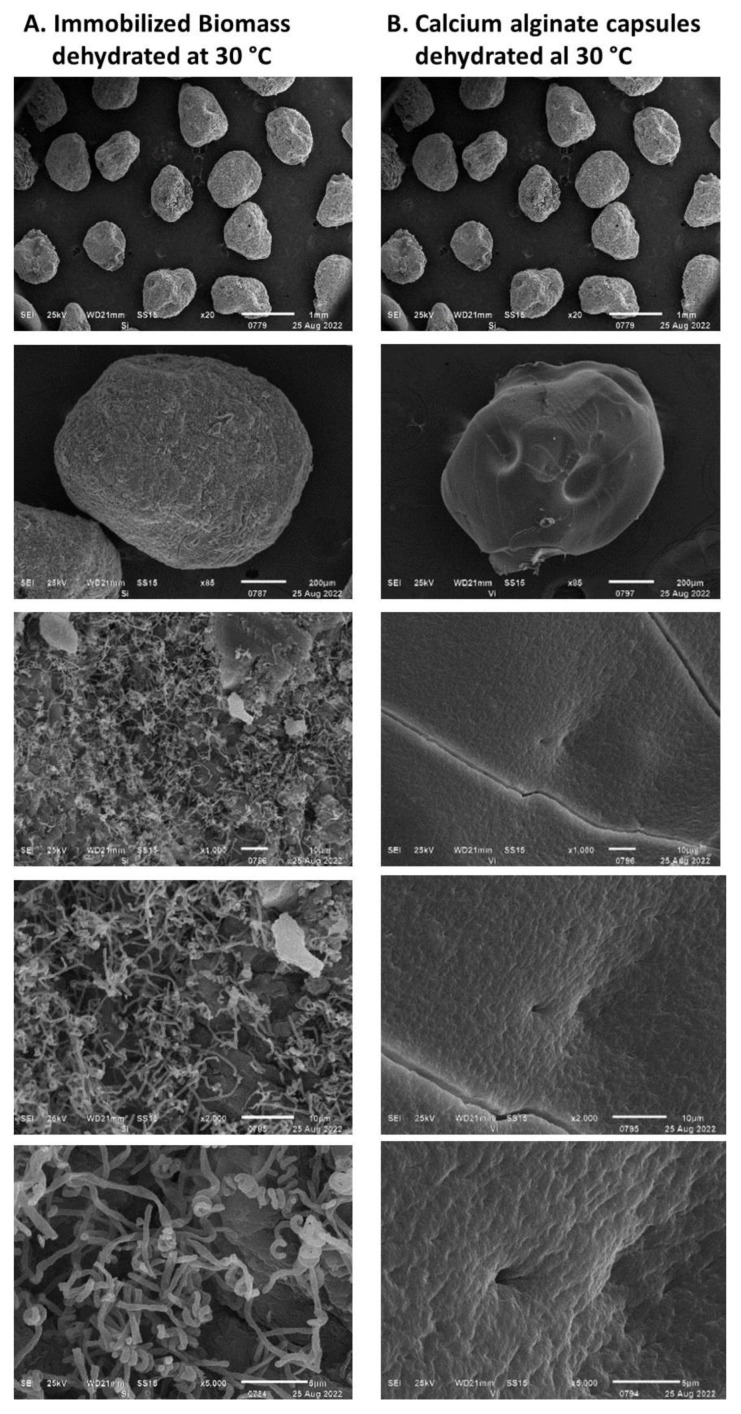
Calcium alginate capsules with (**A**) and without (**B**) biomass dehydrated with heat at 30 °C.

**Figure 14 polymers-15-00207-f014:**
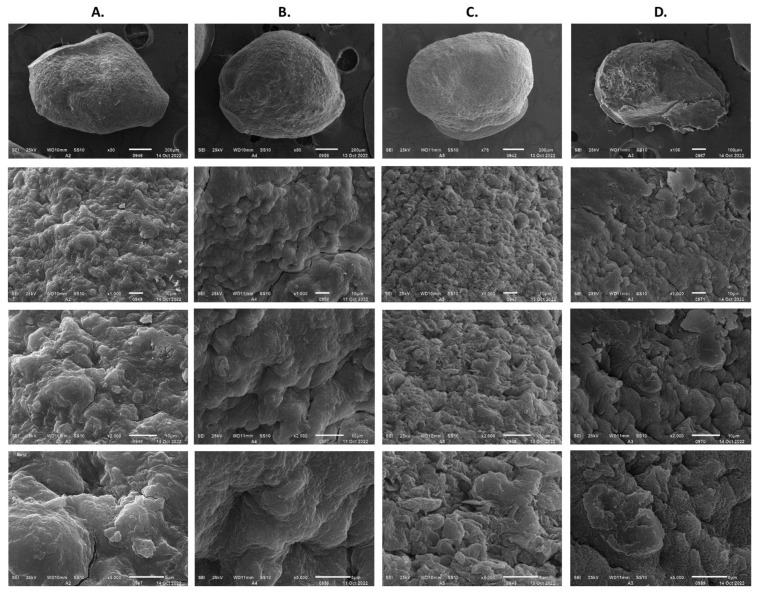
Calcium alginate capsules including protectors: (**A**) Trehalose incorporated in the YGM medium for the Streptomyces propagation. (**B**) Trehalose incorporated in the sodium alginate solution. (**C**) Kaolin incorporated in the sodium alginate solution. (**D**) Gum arabic incorporated into the alginate capsule surface.

**Table 1 polymers-15-00207-t001:** Viability (%) of the *Streptomyces* CDBB1232 mycelium after its immobilization in a calcium alginate matrix and subsequent dehydration.

Protector	Included in:	Encapsulation	Dehydration at 4 L min^−1^	Encapsulation and Dehydration
Trehalose * YGM	Culture medium Sodium alginate (1.5%)	100 ± 0 ^a^	10 ± 1 ^cde^	13 ± 3 ^cde^
YGM + Trehalose	Sodium alginate (1.5%)	89 ± 7 ^a^	27 ± 6 ^c^	24 ± 3 ^cd^
YGM + Kaolin	Sodium alginate (1.5%)	63 ± 6 ^b^	4 ± 0.3 ^e^	3 ± 0.1 ^e^
YGM Arabic Gum	Sodium alginate (1.5%) Alginate capsule surface	61 ± 4 ^b^	100 ± 0 ^a^	87 ± 5 ^a^
YGM	Sodium alginate (1.5%)	64 ± 5 ^b^	26 ± 2 ^c^	16 ± 2 ^cde^

* YGM culture medium added with trehalose. It was used in the propagation process of *Streptomyces* CDBB1232. Average values (± SE) that do not share an equal letter (^a–e^) are significantly differ-ent according to Tukey’s test (α = 0.05). Two-way ANOVA: F (support) = 204.62; df = 4; *p* < 0.0001; F (treatment) = 449; df = 2; *p* < 0.0001; F (support-treatment) = 97.56; df = 8; *p* < 0.0001.

## Data Availability

Not applicable.

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
