# Peer review of "Polymeric Encapsulate of Streptomyces Mycelium Resistant to Dehydration with Air Flow at Room Temperature"

_polymers, 2022, doi:10.3390/polym15010207_

Round 1
Reviewer 1 Report
The article " Polymeric encapsulate of Streptomyces mycelium resistant to 3 dehydration with air flow at room temperature" submitted by María Elena Mancera-López , Josefina Barrera-Cortés , Roberto Mendoza Serna , Armando Ariza-Castolo , Rosa Santillan looks pretty good and I do not have any advices anything to change. However what I would like to see in such a paper is a table which compares different peculiarities of the methods already existing and introduced in that paper. If I see such a table I would conclude that it is reasonable to publish this manuscript.
Author Response
The authors thank reviewer 1 for his comments. The suggested table was prepared and included in the resubmitted paper as supplemental Table S1. Two more supplemental tables were included as complement to the first one; the references are included in the main document. In addition, at the suggestion of reviewer 2, the different sections of the manuscript were revised. The corrections made are marked in the updated manuscript.
Author Response
Point 1. The manuscript entitled “Polymeric encapsulate of Streptomyces mycelium resistant to dehydration with air flow at room temperature” is well performed. However, it may need major revision in the entire manuscript.
Response 1: The authors appreciate the comments and corrections suggested by reviewer 2. The manuscript was revised and corrected as described throughout this document.
Point 2. The selection of methodology is very simple. The effect of encapsulation efficiency not only depends on the alginate. Another chelating agent is also required. This is not performed in this methodology.
Response 2: The authors appreciate the reviewer's comments. Effectively, the encapsulation efficiency of biological agents depends on different factors such as the properties of the encapsulating material, the compatibility between the encapsulating materials and the substances to be encapsulated, as well as on the encapsulation method (Schoebitz et al., 2013; Pungrasmi et al., 2019). For encapsulating biological agents such as the different genera of microorganisms (bacteria, fungi, yeasts) in their different stages (vegetative cells (mycelium) and spores), sodium alginate gelled with CaCl2 has been one of the most widely used materials, despite the gel fragility to drying processes (Martínez-Cano et al., 2022; Schoebitz et al., 2013). However, this disadvantage has been overcome by combining or covering the calcium alginate gel with other polymers (Simó et al., 2017; Saberi-Riseh et al., 2021). This strategy has been very successful and therefore it is the strategy that we apply in the preparation of the dried encapsulation of the Streptomyces mycelium, but to reduce the mycelium viability loss.
Simo et al. (2017) reported: “…chelating agents (such as citrate, phosphate, and lactate) or anti-gelling cations (Na+ and Mg2+) frequently presented in biological and food applications, can cause a reduction in alginate gel mechanical stability or even the complete liquefaction of the gel (Ching, Bansal, & Bhandari, 2015).”
Aware of this type of report, the inclusion of chelating agents in the alginate gel was not considered for the Streptomyces mycelium encapsulation. In two reviewed articles, chelating agents have been used to form gels with very particular characteristics (Chueh et al., 2010; Yamamoto et al., 2019).
Regarding the efficiency of encapsulation of gelled alginates with CaCl2, the reported percentages are generally high (Martínez-Cano et al., 2022, Zhansheng et al., 2012).
Considering reviewer 2's comments, the discussion of results related to encapsulation efficiency was expanded. The information was included in lines (568-615) of the updated manuscript.
References
Pungrasmi W, Intarasoontron J, Jongvivatsakul P, Likitlersuang S. Evaluation of Microencapsulation Techniques for MICP Bacterial Spores Applied in Self-Healing Concrete. Sci Rep. 2019, 28; 9(1):12484. DOI: 10.1038/s41598-019-49002-6.
“This study determined the most appropriate technique to encapsulate spores of Bacillus sphaericus LMG 22257 with sodium alginate so as to protect the bacterial spores during the concrete mixing and hardening period. Three techniques (extrusion, spray drying, and freeze drying) to encapsulate the bacterial spores with sodium alginate were evaluated. The freeze-drying process provided the highest bacterial spore survival rate (100%), while the extruded and spray-dried processes had a lower spore survival rate of 93.8% and 79.9%, respectively.”
Martínez-Cano, B.; Mendoza-Meneses, C.J.; García-Trejo, J.F.; Macías-Bobadilla, G.; Aguirre-Becerra, H.; Soto-Zarazúa, G.M.; Feregrino-Pérez, A.A. Review and Perspectives of the Use of Alginate as a Polymer Matrix for Microorganisms Applied in Agro-Industry. Molecules 2022, 27, 4248. https://doi.org/ 10.3390/molecules27134248
“Sodium alginate is a very versatile polysaccharide with many applications due to its high-tech functionality, it is economical to produce, and it can be obtained in bulk; it is not toxic, it has continuos quality, it is practically sterile, and it is biocompatible with microorganisms [1]. It is currently functional in the food, textile, agrotechnological, biomedical, and pharmaceutical industries since it can be easily modified through chemical and physical reactions to create matrices such as hydrogels, microspheres, microcapsules, sponges, and fibers [2].”
“In addition to starch, sodium bentonite has been used in combination with alginate to develop effective biofertilizer formulations that minimize production costs. The mixture encapsulation efficiency is almost 100, 88.9% of Raoultella planticola bacteria survived after 6 months of storage and swelling, biodegradability, and release rate were found to increase with increasing alginate content, presenting a first-order release, which proves a slow-release, ideal for farmland [115]”
“There are different methods for microencapsulation with alginate: extrusion, which is easy to apply and industrialize but is not compatible with thermosensitive microorganisms; coextrusion, which is a process for a large payload but has high costs; spray-drying, it is an economical technology and equipment is widely available, it has good encapsulation efficiency, and it is a fast process, but it has a low production volume; spray-cooling is an economical technology for encapsulation but requires high energy and a long process time, which makes it more expensive; and complex coacervation, it has a high payload and no specific equipment is required; however, it is a costly process due to the complexity of the technique [138].”
Schoebitz, M.; López, M.D.; Roldán, A. Bioencapsulation of microbial inoculants for better soil–plant fertilization: A review. Agron. Sustain. Dev. 2013, 33, 751–765.
“Numerous advantages related to the bioencapsulation of rhizobacteria are found, for instance, controlled release of bacteria into the soil, protection of microorganisms in the soil against biotic and abiotic stresses, and contamination reduction during storage and transport.”
“Sodium alginate is one of the most commonly used products for the bioencapsulation of microorganisms. The resulting inoculum is used for various purposes: the immobilization of bacteria (Bashan 1986; Bashan et al. 2002), fermentation and application of biological control agents (Bashan and Holguin 1994), or biostimulants for plant growth (Bashan and Levanony 1990; Schoebitz et al. 2012)”
“The properties of alginate are variable according to the origin of the seaweed and the manufacturing process. For instance, in relation to their molecular weight, alginates will have different solubility properties and complexation with calcium”
Saberi-Riseh, R.; Moradi-Pour, M.; Mohammadinejad, R.; Kumar, V. Biopolymers for Biological Control of Plant Pathogens: Advances in Microencapsulation of Beneficial Microorganisms. Polymers 2021, 13, 1938.
“ The disadvantages of alginate beads are that they are not compatible with alkaline conditions [88]. However, the faults of this polymer can be atoned by mixing alginate with various biopolymer, and coating the capsules with another polymer such as gelatin [89] and chitosan, for example in (Figure 7a) was demonstrated more details about this process, where primary microcapsules produced by alginate was consecutively coated by chitosan [50]. Therefore, as stated that strong ionic interactions between the anionic group (alginate) and cationic group (chitosan) (showed in Figure 7b), cause improved effective protection and capsule stability.“
Łętocha, A.; Miastkowska, M.; Sikora, E. Preparation and Characteristics of Alginate Microparticles for Food, Pharmaceutical and Cosmetic Applications. Polymers 2022, 14, 3834. https://doi.org/10.3390/ polym14183834
“The interaction of alginates with divalent cations, which are in particular calcium cations, leads to the formation of biodegradable gels [38]. Polymerization is based on the cross-linking of copolymers through ionic bonds between Ca2+ cations and alginate anions [38,44].”
Zhansheng Wu, Lina Guo, Shaohua Qin, Chun Li. Encapsulation of R. planticola Rs-2 from alginate-starch-bentonite and its controlled release and swelling behavior under simulated soil conditions. J Ind Microbiol Biotechnol (2012) 39:317–327. DOI 10.1007/s10295-011-1028-2
“The plant growth-promoting bacteria (PGPR) Raoultella planticola Rs-2 was encapsulated with the various blends of alginate, starch, and bentonite for development of controlled-release formulations. The stability and release characteristics of these different capsule formulations were evaluated. The entrapment efficiency of Rs-2 in the beads (capsules) was more than 99%.”
“Encapsulation of bacteria within beads was carried out under sterile conditions. Encapsulated cell formulations were obtained by the extrusion technique using a method similar to that described by Wu et al. [27].”
Simó, G.; Fernández‐Fernández, E.; Vila‐Crespo, J.; Ruipérez, VV.; Rodríguez‐Nogales, J.M. Research progress in coating techniques of alginate gel polymer for cell encapsulation. Carbohydrate Polymers, 2017, 170, 1-14, https://doi.org/10.1016/j.carbpol.2017.04.013.
Chueh BH, Zheng Y, Torisawa YS, Hsiao AY, Ge C, Hsiong S, Huebsch N, Franceschi R, Mooney DJ, Takayama S. Patterning alginate hydrogels using light-directed release of caged calcium in a microfluidic device. Biomed Microdevices. 2010; 12(1):145-51. DOI: 10.1007/s10544-009-9369-6.
Yamamoto, K.; Yuguchi, Y.; Stokke, B.T.; Sikorski, P.; Bassett, D.C. Local Structure of Ca2+ Alginate Hydrogels Gelled via Competitive Ligand Exchange and Measured by Small Angle X-Ray Scattering. Gels 2019, 5, 3; doi:10.3390/gels5010003
Point 3. The test sample is a dried one. For most of the analysis is carried out in the sample as a solution form. How it is possible?
Response 3: The authors appreciate reviewer 2's observation. We place the question in section 2.4 (Dehydration process of encapsulated biomass). The experimental procedure related to the microbial count was reviewed and corrected, see sections 2.2, 2.3, and 2.4. The following diagram shows the flow of biological material (wet or suspended biomass) in the applied experimental procedure. After encapsulation, the biomass in suspension has the purpose of analyzing the microbial count
Point 4. Include the name and kind of the alginate in the abstract and materials, also where’s materials section.
Response 4: The authors appreciate the reviewer's suggestion. Information on the type of sodium alginate used in the present study was included both in the abstract and in the materials and methods section. This information is marked in red in the corrected document, specifically, it is located on lines 18-22 and 174-176.
Point 5. The introduction needs to improve, need to enhance the research gap between previous studies and this study. Check the unit as per author guidelines. It is mismatched in the manuscript?
Response 4: The authors appreciate the reviewer's suggestion. The introduction was revised, and information on the encapsulation of biological material was included. A table containing information on encapsulated formulations based on strains of the genus Streptomyces was prepared. The Table was included as supplementary material.
Point 6. Grammatical mistakes are majorly presented. Check the grammar mistake in the entire manuscript.
Response 6: The authors appreciate the reviewer's observation. The corrected manuscript was submitted for review of English grammar.
Point 7. Methodology protocol is not well performed?
Response 7: The authors are grateful for the comments of reviewer 2. To visualize the experimental methodology, a flowchart of the experimental procedure was produced. The figure was included as supplementary material. In addition, the introduction and materials and methods sections were revised. The changes made were marked in the corrected document. (e.g. lines 229-232)
Point 8. Also, are some other ingredients also required for enhancing the EE of the developed product?
Response 8: The authors appreciate the reviewer's question. According to what was reported by Martínez-Cano et al. 2022 and Zhansheng et al, 2012, the efficiency of microbial cells encapsulated in calcium alginate is high (see the answer to point 2). However, we consider that the strain propagation method and its physiological state, as well as the encapsulation method, are factors that determine the viability of the strain. In the present study, the propagation of Streptomyces with trehalose as a carbon source generated viability percentages of 100% of the encapsulated strain. However, when using only glucose as a carbon source (YGM medium), the viability was 64%. The ANOVA of the mycelium viability as a function of the encapsulating support (sodium alginate solubilized in YGM medium, sodium alginate solubilized in YGM medium and added with kaolin, alginate capsules covered with Arabic gum), did not show significant differences in the viability of the encapsulated mycelium. The application of Arabic gum to the capsules prepared with the mycelium grown in trehalose and glucose could have prevented its loss of viability during the drying treatment; however, this option would no longer have an impact on the encapsulation efficiency, but as a protective means in the drying process.
Point 9. The result and discussion part are getting boring. Need to improve.
Response 9: The authors appreciate the reviewer's observation. The results and results discussion sections were revised. The most important changes can be seen in the results discussion section. The changes are marked in the updated manuscript.
Point 10. The grammatical mistake majorly happened in the entire manuscript.
Response 10: The authors appreciate the reviewer's suggestion. The corrected manuscript was submitted for review of English grammar.
Point 11. The references should be updated.
Response 11:
The authors appreciate the reviewer's suggestion. A bibliographic review was carried out and some references were changed for more current information, basically from the year 2022.
Point 12. Font in figure 1, it’s not easy to read.
Response 12: The authors appreciate the reviewer's observation. The font size in the figure was increased.
Point 13. Where’s the characterization of capsulation (zeta potential and encapsulation efficiency)
Response 13: The authors appreciate the editor's observation. According to the consulted literature, the Z potential can be measured by DLS and in particles with a diameter of microns (10) and nanometers. In the present work, the capsule diameter was 3 mm and the Zetasizer ZS90 equipment (Malvern Instruments, Worcestershire, U.K.) available for this analysis, only allows analyzing particles with a diameter in the range of 0.3 nm to 10 microns. For this reason, the Z potential of the produced capsules was not measured.
According to Morais et al. (2020), the efficiency of cell encapsulation increases with the negative Z potential of the calcium alginate. Other authors have reported the Z potential of alginate particles by the effect of pH variation, alginate concentration, the concentration of the gelling substance (CaCl2), the diameter of the alginate capsule, and due to the effect of mixing the alginate with other polymers or including additives [Barros Silverio et al., 2018; Wang et al., 2021]. The fact of containing biomass also changes the Z potential of the alginate matrix. The value of the Z potential determined in calcium alginate gels is around -70 mV. High concentrations of CACl2 can produce positive values of the Z potential [Barros Silverio et al., 2018]. Alginate beads with biomass have given low values of -2 and -9 mV.
The calculation of the encapsulation efficiency requires the determination of the remaining microbial count in the gelling solution (CaCl2, 0.2M). But having encapsulated mycelium and with knowledge of its fragility, in the present work we focus on determining the viability of the mycelium. However, the high viability percentages obtained from the encapsulation of the Streptomyces mycelium propagated with trehalose as culture medium, allows us to assume that the sodium alginate used is adequate to encapsulate the mycelium of the studied Streptomyces strain.
References:
Morais, E.C.; Schroeder, H.T.; Souza, C.S.; Rodrigues, S.R.; Rodrigues, M.I.L.; Homem de Bittencourt Jr., P.I.; Dos Santos J.H.Z. Comparative study on the influence of the content and functionalization of alginate matrices on K-562 cell viability and differentiation. Journal of Materials Research 2020, 35(10), 1249-1261. DOI: 10.1557/jmr.2020.96
Barros Silverio, G.; Sakanaka, L.S.; Dutra Alvim, I.; Shirai, M.A.; Ferreira Grosso, C.R. Production and characterization of alginate microparticles obtained by ionic gelation and electrostatic adsorption of concentrated soy protein. Ciência Rural, Santa Maria, 2018, 48:12, e20180637. https://doi.org/10.1590/0103-8478cr20180637
Wang, Y.; Tan, C.; Davachi, S.M.; Li, P.; Davidowsky, P.; Yan B. Development of Microcapsules Using Chitosan and Alginate via W/O Emulsion for the Protection of Hydrophilic Compounds by Comparing with Hydrogel Beads. International Journal of Biological Macromolecules 2021, 177, 92–99. https://doi.org/10.1016/j.ijbiomac.2021.02.089
Point 14. Line 101 (Mexico City, Mexico)
Response 14: The authors appreciate the reviewer's observation. Done.
Point 15. Figure 5 and 6, the statistical analysis should be written like figures 9 and 10
Response 15: The authors appreciate the reviewer's observation. At the reviewer's suggestion, the ANOVA of the experimental data was included in the corresponding figure captions.
Point 16. Table 1 should be written standard deviation under the table.
Response 16: The authors appreciate the reviewer's observation. At the reviewer's suggestion, the ANOVA of the experimental data was included under the table.

Round 2
Reviewer 2 Report
Accept